# RETHINKING *ab initio* NEURAL WAVE FUNCTION: GNNS AT ELECTRON LEVEL

## ABSTRACT

Solving the many-body Schrödinger equation remains one of the most fundamental yet challenging problems in quantum mechanics. Recent advances suggest that neural networks can serve as expressive and powerful ansätze; however, a key opportunity remains in elucidating a clear, intuitive bridge between the network architecture and the underlying physics, especially for a broader AI audience. In this work, we rethink the design of Neural Wave Functions (NWFs) by leveraging the inherently graphical structure of many-body systems. We propose the Graph Neural Wave Function (GNWF), a novel approach that leverages graph neural networks (GNNs) to represent quantum wave functions, inspired by the intrinsically graphical nature of many-body systems. By integrating GNWF with variational quantum Monte Carlo (VMC), GNWF learns wave functions solely from first principles, without reliance on supervised data or pre-computed targets. To assess its effectiveness, we benchmark GNWF on quantum chemistry systems, achieving comparable accuracy with a 24% speedup over recent peer NWF: Psiformer. This work effectively extends the application of GNNs beyond atomic representations to the electron level, unlocking new possibilities for machine learning in quantum many-body physics.

## 1 INTRODUCTION

Solving the wave function of particles is one of the most fundamental and challenging problems in both quantum physics and quantum chemistry Szabo & Ostlund (1996). The wave function maps the spatial coordinates of particles to a value representing their probability distribution. However, solving the wave function for a single particle, let alone for multiple particles in a molecular system, is notoriously difficult due to the exponential scaling of the problem Weimer et al. (2021). Accurate solutions are essential for predicting molecular properties, such as energy surfaces and ground state energies. Recent advances have seen the adoption of neural networks to approximate the wave function, known as neural wave functions (NWFs) Carleo & Troyer (2017); Pfau et al. (2020); von Glehn et al. (2022); Hermann et al. (2020), which have demonstrated promising accuracy in predicting energies with chemical precision Gerard et al. (2022); von Glehn et al. (2022). Nevertheless, the development of more efficient and physically grounded NWF models remains an open challenge.

In this paper, we propose a novel approach that leverages graph neural networks (GNNs) for learning wave functions. Given the 3D structure of molecules, GNNs have been successfully applied to predict molecular properties Schütt et al. (2018); Gasteiger et al. (2020); Liu et al. (2021). However, traditional supervised learning approaches that focus on atomic-level representations have inherent difficulty in capturing the complex interactions among electrons, nuclei, and their positions, which significantly influence the system's Hamiltonian and correlated energies Reiser et al. (2022). To address this, we propose to treat electrons and nuclei as graph nodes, integrating GNNs with the *ab initio* method, variational Monte Carlo (VMC) Foulkes et al. (2001), to (hopefully) enable first-principles predictions. Unlike previous NWF models that adopt multilayer perceptrons (MLPs) or transformers to process particle features, our GNN-based ansätz may provide a more transparent flow of information between electrons and nuclei. Additionally, the use of GNNs helps reduce redundant edges between nodes, leading to fewer parameters and a more efficient training process.

We validate the performance of the proposed Graph Neural Wave Function (GNWF) on a range of quantum chemistry benchmark systems, including both atoms and molecules (up to 30 electrons). Our results show that GNWF outperforms existing methods by providing competitive accuracy in

ground state energy calculations while accelerating the training process of the SOTA variational Monte Carlo-based NWF models. To further demonstrate the stability of GNWF, we also examine the energy surface of water with varying bond distances and angles. **The highlights of this paper are:**

- We revisit the NWF ansatz and propose an *ab initio* graph neural wave function model inspired by the intrinsic graphical structure of electronic systems. Specifically, we combine GNNs with VMC to construct many-body wave functions based on learned node features.
- A carefully designed paradigm preserving the indexing invariant and symmetric nature of quantum many-body systems. GNNs simplify the message passing scheme in existing NWFs Pfau et al. (2020); von Glehn et al. (2022), resulting in fewer parameters and faster training.
- urate calculations of the ground state energy within chemical accuracy for a variety of atoms and molecules. The approximation of the energy surface of water demonstrates the performance stability of GNWF across different bond distances and angles.

## 2 PRELIMINARIES AND RELATED WORKS

### 2.1 QUANTUM MECHANISMS

We start with a brief introduction to the Schrödinger equation, which describes the physical properties of a quantum system. For its tractability for existing learning technology, we follow the literature and consider the time-independent Schrödinger equation can be written as:

$$\hat{H}\,\psi(\mathbf{pos}) = E\,\psi(\mathbf{pos}), \tag{1}$$

where $E$ is a scalar denoting the energy, $\hat{H}$ is an Hermitian operator satisfying $\hat{H}^\dagger = \hat{H}$ called the (problem) Hamiltonian. $\psi(\mathbf{pos})$ is the wave function that maps the spacial coordinates of $n$ particles $\mathbf{pos} \in \mathbb{R}^{3n}$ to $\mathbb{C}$, where $|\psi(\mathbf{pos})|^2$ serves as the probability. For the considered time-independent Schrödinger equation here, which is an eigenfunction, we can solve the minimum eigenvalue, i.e. the ground state energy of the system, with variational principle:

$$E = \frac{\langle\psi|\hat{H}|\psi\rangle}{\langle\psi|\psi\rangle} \geq E_0, \tag{2}$$

where $E_0$ is the ground state energy. Note that the problem Hamiltonian $\hat{H}$ contains all the system information. For molecular systems, $\hat{H}$ under the Born-Oppenheimer (BO) approximation is given by Foulkes et al. (2001):

$$\hat{H} = -\frac{1}{2}\sum_i \nabla_i^2 + \sum_{i>j} \frac{1}{|\mathbf{pos}_i - \mathbf{pos}_j|} - \sum_{iI} \frac{Z_I}{|\mathbf{pos}_i - \mathbf{R}_I|} + \sum_{I>J} \frac{Z_I Z_J}{|\mathbf{R}_I - \mathbf{R}_J|}. \tag{3}$$

The BO approximation fixes the nuclei at position $\mathbf{R}$ while the electrons are dynamic with positions $\mathbf{pos}$. The key to solving the ground state of a quantum system is to get an accurate approximation of the wave function (namely ansätz), and the fundamental methods should be able to obtain the wave function only from the atomic positions, which are called *ab initio* Martin (2020), indicating they are calculating from first principle. A widely used *ab initio* approach to construct neural wave functions is Variational Monte Carlo (VMC) Foulkes et al. (2001). For an $N$-electron system, the coordinates of all particles are collectively denoted by $\mathbf{pos} = (\mathbf{pos}_1, \mathbf{pos}_2, \ldots, \mathbf{pos}_N)$, with the parameterized many-body wave function as $\psi_\theta(\mathbf{pos})$ which is parameterized by $\theta$. The energy expectation value, which serves as the loss function for VMC optimization, is expressed as:

$$\mathcal{L}(\theta) = \frac{\langle\psi_\theta|\hat{H}|\psi_\theta\rangle}{\langle\psi_\theta|\psi_\theta\rangle} = \mathbb{E}_{\mathbf{pos}\sim|\psi_\theta(\mathbf{pos})|^2}\left[E_{\mathrm{L}}(\mathbf{pos})\right], \tag{4}$$

where the local energy is defined as $E_{\mathrm{L}}(\mathbf{pos}) = \frac{\hat{H}\psi_\theta(\mathbf{pos})}{\psi_\theta(\mathbf{pos})}$. Monte Carlo integration, often using the Markov Chain Monte Carlo (MCMC) method, facilitates efficient sampling over the high-dimensional configuration space. The optimization involves iterative updates of the parameters $\theta$ using the gradient of the loss with learning rate $\lambda$: $\theta_{t+1} = \theta_t - \lambda\nabla_\theta\mathcal{L}(\theta_t)$, Convergence is achieved when the loss $\mathcal{L}(\theta)$ reaches a minimum, and its variance approaches zero, yielding the ground-state wave function $\psi_\theta(\mathbf{pos})$. For electronic systems, which follow the Pauli exclusion principle, have an antisymmetric

Figure 1: **Schematic pipeline of the proposed GNWF.** The process begins with the nuclei positions and an initial trial wave function. Electrons whose positions are sampled from the trial wave function and fixed nuclei form a 3D graph, where node and edge embeddings are initialized with selected features. Message passing occurs exclusively between electron-nucleus pairs, with each node being updated via a weighted sum of its neighboring node features and edge features. The weights decay exponentially with distance (illustrated by the edge thickness in the figure). The electron features are then treated as the single-particle wave function, and the final GNWF is constructed by combining these features through a Slater determinant.

nature imposing specific constraints on the wave function that exchanging two particles will lead to a sign change:

$$\psi(\ldots, \mathbf{pos}_i, \ldots, \mathbf{pos}_j, \ldots) = -\psi(\ldots, \mathbf{pos}_j, \ldots, \mathbf{pos}_i, \ldots). \tag{5}$$

To ensure these constraints, one can adopt the well-studied Slater determinant to construct many-body wave functions from single-particle ones $\phi_i(\mathbf{pos}_j)$ Foulkes et al. (2001):

$$D(\mathbf{pos}) = \det\left(\phi_i(\mathbf{pos}_j)\right). \tag{6}$$

## 2.2 NEURAL NETWORK QUANTUM MONTE CARLO

In recent years, researchers have been increasingly incorporating neural networks into quantum Monte Carlo (QMC) methods Foulkes et al. (2001), particularly Variational Monte Carlo (VMC) and Diffusion Monte Carlo (DMC). This approach, termed Neural Network Quantum Monte Carlo (NNQMC) Hermann et al. (2023), leverages neural networks as wavefunction ansätz. Compared to traditional parametric analytical function ansätzes, neural networks offer superior expressiveness, scalability, and computational efficiency. These advantages enable NNQMC to achieve remarkable performance on a wide range of problems, in some cases surpassing the accuracy of the "gold standard" CCSD(T) method von Glehn et al. (2022).

Graph Neural Networks were first introduced into VMC by Yang et al. (2020) for the lattice problems and further extended in Roth & MacDonald (2021); Scherbela et al. (2024). Regarding neural architectures for NNQMC, GNNs are introduced via PESNet Gao & Günnemann (2021), later extended to PESNet++ Gao & Günnemann (2022), Globe+Moon Gao & Günnemann (2023), among others Gao & Günnemann (2024). However, these methods take GNNs as a supervised complement for other NWFs (e.g. Pfau et al. (2020)), which neglects the fact that NWFs are designed to be *ab initio* (without empirical parameters or experimental data). In this paper, we would like to demonstrate that GNN, capturing the structures of electrons and nuclei, provides an effective message passing scheme for learning wave functions from first principles.

## 3 METHODOLOGY

We will introduce the proposed graph neural wave function (GNWF). The scheme of the proposed GNWF is shown in Fig. 1 with a pseudo-code in Alg. 1. The algorithm starts with the trial wave function as the Hartree-Fock state for the MCMC to sample electron positions. These sampled positions are then utilized to construct embeddings.

## 3.1 INITIAL EMBEDDING

Naturally, electrons and nuclei form a fully connected 3D graph with interaction-dependent connectivity. To reduce the cost of message passing, we eliminate intra-electron and intra-nucleus edges, resulting in a reduced graph $\mathcal{G}$ with node set $\mathcal{V} = \mathcal{V}_{ele} \cup \mathcal{V}_{nuc}$ and edge set $\mathcal{E} = \{e_{i,j}\},\ i \in \mathcal{V}_{ele}, j \in \mathcal{V}_{nuc}$. In this framework, electron-electron correlations are not modeled by direct edges but are emergent

properties learned through the shared interactions with nuclei. Each nucleus and electron carries fundamental properties dictated by the Hamiltonian (Eq. 3):

1) Nuclei: Spatial coordinates $\mathbf{pos} \in \mathbb{R}^3$, charge $c \in \mathbb{Z}^+$.

2) Electrons: Spatial coordinates $\mathbf{pos} \in \mathbb{R}^3$, unit charge $c = -1$, and spin (spin-up $\uparrow$ as 1 and spin-down $\downarrow$ as $-1$).

To compensate for the edge reduction, we introduce two additional structural features: the mean-field representation of electron interactions and the center of charge. The mean-field representation for electron $i$ is

$$\mathbf{e2c}_i = \sum_{j \in \mathcal{V}_{ele}} (\mathbf{pos}_j - \mathbf{pos}_i), \tag{7}$$

where $\mathbf{e2c}_i \in \mathbb{R}^3$. The center of charge $\mathbf{charge\_cen} \in \mathbb{R}^3$ is defined as:

$$\mathbf{charge\_cen} = \frac{1}{|\mathcal{V}|} \sum_{i \in \mathcal{V}} |c_i| \mathbf{pos}_i, \tag{8}$$

where $|\mathcal{V}|$ denotes the number of nodes, $|c_i|$ is the absolute value of the charge of node $i$. For each nucleus, we have:

$$\mathbf{a2c}_j = \mathbf{pos}_j - \mathbf{charge\_cen}. \tag{9}$$

The addition of $\mathbf{a2c}$ ensures nuclei embeddings are dynamic under Monte Carlo sampling, resolving the problem that the position and charge of the nucleus are constant under the Born-Oppenheimer approximation. Thus, each nucleus is initialized with a 7-dimensional embedding, and each electron with an 8-dimensional embedding. Each nucleus-electron edge is encoded as a 3D relative position vector:

$$\mathbf{emb\_edge}_{ij} = \mathbf{pos}_j - \mathbf{pos}_i. \tag{10}$$

Unlike other GNN methods for molecular learning, the *ab initio* GNN wave function does not need $SE(3)$ invariant and equivariant for initial embedding. The learned wave function will rotate and translate with the coordinates of nuclei, and thus, we only need to ensure the index of nodes does not affect the results during initial embedding and further aggregation. This is why we use summation in Eq. 7 and Eq. 8 instead of concatenating the embedding of related particles as in Pfau et al. (2020); von Glehn et al. (2022), where changing the node index will alter the node embedding.

The initial embeddings for electrons, nuclei and edges are then projected into a fixed hidden dimension with a linear layer. We denote the feature representations as $\mathbf{he}$ (electrons), $\mathbf{ha}$ (nuclei), and $\mathbf{g}$ (edges). The initial features at layer 0 are denoted as:

$$\begin{aligned}
\mathbf{ha}^0 &= \tanh(\mathbf{W}_a)\mathbf{emb\_nuc}) + \mathbf{b}_a, \\
\mathbf{he}^0 &= \tanh((\mathbf{W}_e)\mathbf{emb\_ele}) + \mathbf{b}_e, \\
\mathbf{g}^0 &= \tanh((\mathbf{W}_g)\mathbf{emb\_edge}) + \mathbf{b}_g.
\end{aligned} \tag{11}$$

**On the Absence of Electron-Electron Edges:** Our graph construction intentionally omits direct edges between electrons (E-E edges), a design choice motivated by both computational and physical considerations. Within the VMC framework, electron positions are stochastic samples; thus, modeling interactions via instantaneous positional edges may not accurately represent the underlying wave-function correlations. Instead, E-E interactions are captured indirectly—and effectively—through message passing mediated by nuclei. After two layers, information can propagate from one electron to another via their shared connections to nuclei, enabling the model to learn these crucial correlations.

This approach aligns with a physical intuition: the repeated sampling of an electron's position effectively forms an orbital centered around nuclei. While the intra-orbital electron-nucleus distance remains relatively stable, the inter-electron distances across different orbitals vary significantly during sampling. Focusing on electron-nucleus interactions provides a more stable foundation for modeling the system's average behavior.

Notably, a similar philosophy of decoupling direct E-E interaction from the primary network architecture is employed in state-of-the-art models like Psiformer von Glehn et al. (2022), which relegates this complex interaction to a final Jastrow factor. Our results and ablations confirm that this simplified graph structure, while reducing computational complexity, remains sufficient for achieving high accuracy, demonstrating that E-E correlations can be effectively learned without explicit E-E edges.

## 3.2 FEATURE AGGREGATION

We now introduce the message passing scheme of the proposed GNNs. Since electrons and nuclei have different characters, we apply two independent networks to update the features. For nucleus $i$, we denote the neighboring node set by $\mathcal{N}_i = \mathcal{V}_{ele}$. The features on these electrons are first concatenated with those on the corresponding edges. Notice that the interaction between two particles decreases with the distance, so we add a weighted factor $\alpha_{ij}$ corresponding to edge $e_{ij}$, defined as:

$$\alpha_{ij} = \exp(-r_{ij}), \tag{12}$$

where $r_{ij} = \|\mathbf{pos}_j - \mathbf{pos}_i\|_2$, and $\| \cdot \|_2$ denotes the Euclidean norm. It fulfills the effect of neighboring particles has an exponential decay with the distance. The weighted aggregation of neighboring features for nucleus $i$ is

$$\sum_{j \in \mathcal{V}_{ele}} \alpha_{ij} \, \text{concat}(\mathbf{he}_j^l, \mathbf{g}_{ij}^l), \tag{13}$$

where $l$ denotes the GNN layer. The feature vector of nucleus $i$ is then concatenated with the neighboring features to form a temporary feature vector:

$$\mathbf{fa}_i^l \leftarrow \text{concat}\Big(\mathbf{ha}_i^l, \sum_{j \in \mathcal{V}_{ele}} \alpha_{ij} \, \text{concat}(\mathbf{he}_j^l, \mathbf{g}_{ij}^l)\Big). \tag{14}$$

Similarly, we construct the temporary feature vector for electron $i$ by aggregating the features of all the nuclei

$$\mathbf{fe}_i^l \leftarrow \text{concat}\Big(\mathbf{he}_i^l, \sum_{j \in \mathcal{V}_{nuc}} \alpha_{ij} \, \text{concat}(\mathbf{ha}_j^l, \mathbf{g}_{ij}^l)\Big). \tag{15}$$

Notice that this message passing scheme ensures the node features for both nuclei and electrons are still indexing invariant. The temporal features then undergo residual updates via learnable transformations:

$$\begin{aligned}
\mathbf{he}^{l+1} &= \tanh(\mathbf{W}_{ae}^l \mathbf{fe}^l + \mathbf{b}_{ae}^l) + \mathbf{he}^l, \\
\mathbf{ha}^{l+1} &= \tanh(\mathbf{W}_{ea}^l \mathbf{fa}^l + \mathbf{b}_{ea}^l) + \mathbf{ha}^l, \\
\mathbf{g}^{l+1} &= \tanh(\mathbf{W}_g^l \mathbf{g}^l + \mathbf{b}_g^l) + \mathbf{g}^l.
\end{aligned} \tag{16}$$

After $L$ layers, the final electron feature $\mathbf{he}$ serves as the learned single-particle wave function representations. The feature initialization and updating procedures of the proposed GNWF are illustrated in Algorithm 1.

## 3.3 CONVERTING FEATURES TO WAVE FUNCTION

Following Pfau et al. (2020); von Glehn et al. (2022), we construct the many-body wave function utilizing the slater determinant from these learned electron features $\mathbf{he}^L$. Given $n = |\mathcal{V}_{ele}|$ number of electrons and $n$ orbitals, the wave function is represented by an $n \times n$ matrix $\mathbf{D}$ with each element $\mathbf{D}_{ij} = \phi_i(\mathbf{pos}_j)$, where $\phi_i$ is the single particle wave function of electron $i$ evaluated at the electron's position $\mathbf{pos}_j$. Therefore, each element of the matrix $\mathbf{D}_{ij}$ involves two electrons $i$ and $j$, and their corresponding features. We utilize a final MLP projection to obtain the value

$$\mathbf{D}_{ij} = \alpha_{ij} \times (\mathbf{W}_i \mathbf{he}_j^L + \beta_i), \tag{17}$$

where $\beta_i$ denotes the bias. $\alpha_{ij}$ is calculated as in Eq. 12 with $i, j \in \mathcal{V}_{ele}$ serves as the exponentially decaying envelope here. $\mathbf{W}_i$ is the trainable weights corresponding to electron $i$ and construct the $i$-th row of the slater determinant $\mathbf{D}$ by integrating with different electron feature $\mathbf{he}_j^L$.

To further enhance expressivity, we also incorporate a multi-head mechanism here to construct $K$ multiple Slater determinants with independent $\mathbf{W}_i^k$ and take the weighted sum of these $K$ determinants as the final output, following Pfau et al. (2020); von Glehn et al. (2022). The calculation of the loss function and parameters is in Section 2.

## 3.4 RETHINKING PREVAILING NWFS

In existing NWF, the determinant is composed of $\mathbf{D}_{ij} = \phi_i(\mathbf{pos}_j) = \alpha_{ij}\mathbf{W}_i\mathbf{he}_j + \alpha_{ij}\beta_i$. The final feature of node $j$ can be seen as a non-linear mapping from $\mathbf{pos}_j$, which is $\mathbf{he}_j^L = \sigma_j\mathbf{pos}$.

---

**Algorithm 1** Graph Feature Initialization and Updating

---

**Input:** graph $\mathcal{G}(\mathcal{V}, \mathcal{E})$, $\mathbf{pos}_i \in \mathbb{R}^3$, charge $c_i \in \mathbb{Z}$, $spin_i \in \{-1, +1\}$, $\forall i \in \mathcal{V}$
$r_{ij} \leftarrow ||\mathbf{pos}_j - \mathbf{pos}_i||_2$
$\alpha_{ij} \leftarrow \exp(-r_{ij})$
$\mathbf{charge\_cen} \leftarrow \frac{1}{|\mathcal{V}|} \sum_{i \in \mathcal{V}} |c_i| \mathbf{pos}_i$
$\mathbf{a2c}_i \leftarrow \mathbf{pos}_i - \mathbf{charge\_cen}$
$\mathbf{e2c}_i \leftarrow \sum_{j \in \mathcal{V}_{ele}} (\mathbf{pos}_j - \mathbf{pos}_i)$
$\mathbf{emb\_nuc}_i \leftarrow \text{concat}(\mathbf{pos}_i, c_i, \mathbf{a2c})$
$\mathbf{emb\_ele}_i \leftarrow \text{concat}(\mathbf{pos}_i, c_i, spin_i, \mathbf{e2c})$
$\mathbf{emb\_edge}_{ij} \leftarrow \mathbf{pos}_j - \mathbf{pos}_i$
$\mathbf{ha}^0 \leftarrow \tanh(\text{matmul}(\mathbf{W}_a), \mathbf{emb\_nuc}) + \mathbf{b}_a$
$\mathbf{he}^0 \leftarrow \tanh(\text{matmul}(\mathbf{W}_e), \mathbf{emb\_ele}) + \mathbf{b}_e$
$\mathbf{g}^0 \leftarrow \tanh(\text{matmul}(\mathbf{W}_g), \mathbf{emb\_edge}) + \mathbf{b}_g$
**for** layer $l = 0$ **to** $L - 1$ **do**
  **for** $i \in \mathcal{V}_{ele}$ **do**
    $\mathbf{fe}_i^l \leftarrow \text{concat}\left(\mathbf{he}_i^l, \sum_{j \in \mathcal{V}_{nuc}} \alpha_{ij} \text{concat}(\mathbf{ha}_j^l, \mathbf{g}_{ij}^l)\right)$
  **end for**
  **for** $i \in \mathcal{V}_{nuc}$ **do**
    $\mathbf{fa}_i^l \leftarrow \text{concat}\left(\mathbf{ha}_i^l, \sum_{j \in \mathcal{V}_{ele}} \alpha_{ij} \text{concat}(\mathbf{he}_j^l, \mathbf{g}_{ij}^l)\right)$
  **end for**
  $\mathbf{he}^{l+1} = \tanh(\mathbf{W}_{ae}^l \mathbf{fe}^l + \mathbf{b}_{ae}^l) + \mathbf{he}^l$
  $\mathbf{ha}^{l+1} = \tanh(\mathbf{W}_{ea}^l \mathbf{fa}^l + \mathbf{b}_{ea}^l) + \mathbf{ha}^l$
  $\mathbf{g}^{l+1} = \tanh(\mathbf{W}_g^l \mathbf{g}^l + \mathbf{b}_g^l) + \mathbf{g}^l$
**end for**
**Return:** $\mathbf{he}^L$

---

Therefore, $\mathbf{D}_{ij}$ can be further derived as $\mathbf{D}_{ij} = \alpha_{ij} \mathbf{W}_i \mathbf{he}_j + \alpha_{ij} \beta_i = \alpha_{ij} \mathbf{W}_i \sigma_j(\mathbf{pos}_j) + \alpha_{ij} \beta_i$. This formulation fundamentally differs from classical methods, where the single-particle wave function for electron $i$ depends only on $i$ and remains fixed across different electron position inputs. As a result, NWF is not a typical Slater determinant method, as it incorporates additional information about electron $j$ and the interaction between $i$ and $j$ in the single-particle wave function.

In the initial embedding construction, other NWF approaches, such as Ferminet Pfau et al. (2020) and Psiformer von Glehn et al. (2022), use concatenation of relative positions and distances. Ferminet concatenates the relative positions and distances of nuclei to each electron as the single embedding, and the relative positions and distances of electrons to each electron as the double embedding. Psiformer, on the other hand, uses only the single embedding. However, this approach introduces several potential issues: 1) Altering the index of the electrons and nuclei will change the initial embedding; 2) the dimension of the initial embedding increases with the system size and might require a large hidden dimension; 3) in the case of atoms, Psiformer's initial embedding has only four dimensions, which serves as the only input to the Transformer model.

There have also been works Gao & Günnemann (2022; 2023; 2024) that utilize GNNs to address the geometric generalization problem, where they aim to train a single model that can produce wavefunctions for a continuous range of molecular geometries. In this paradigm, the GNN acts as a meta-model or a reparameterization tool, whose primary function is to map nuclear coordinates to the parameters of another wavefunction ansatz (e.g., a FermiNet-style network). The GNN itself is not the primary wavefunction ansatz, which is distinct from ou proposed model.

## 4 NUMERICAL EXPERIMENTS

For experiments, the protocols follow Pfau et al. (2020), with the task of ground state energy estimation, on both atoms and molecules, as well as the specific problem of energy surface solving for water. All experiments are conducted on a machine with 1TB memory, 144 cores Intel Xeon Platinum 8352V CPU, and 8 GPUs (NV GeForce RTX 4090). Source code is developed in JAX and will be made publicly available

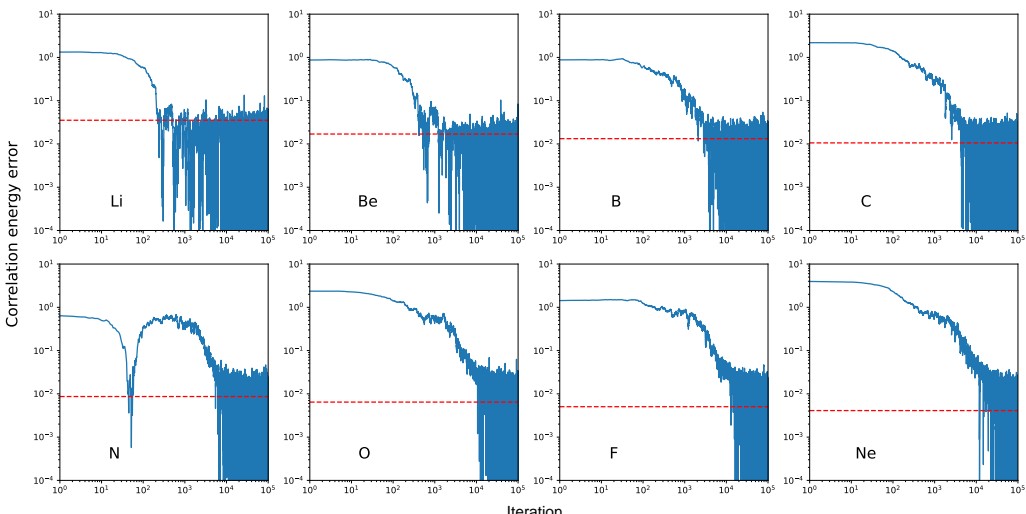

Figure 2: **Results for the second row atoms.** The x-axis is the logarithm of the iterations and the y-axis is the correlation energy error of the calculated energy and the Hartree-Fock state. The red dotted line represents the chemical accuracy ($1.6 \times 10^{-3}$Ha) so that energy below this line would be a chemically meaningful approximation Bogojeski et al. (2020).

### 4.1 SOLVING GROUND STATE ENERGY FOR ATOMS

We first examine the results on the atoms on the second row of the periodic table, including {Li, Be, P, C, N, O, F, Ne} with the number of electrons varying from 3 to 10. These atoms are trivial examples to demonstrate the training curves of the proposed GNWF w.r.t. iterations. Following the experimental setting as in Pfau et al. (2020), we first take 20K iterations of pre-train with the Hartree-Fock method and then conduct 100K training iterations for the GNWF. Since we are pre-training the model with the Hartree-Fock method, the final energy of the proposed method should be smaller than the energy of the Hatree-Fock method. Denote the exact energy as $E_{exact}$ and the energy of the Hartree-Fock method as $E_{HF}$, the correlation energy error of the energy $\hat{E}$ is calculated by $\frac{\hat{E}-E_{HF}}{E_{exact}-E_{HF}}$.

Figure 2 reports the correlation energy error w.r.t. the iterations of all eight atoms. Since the proposed GNWF excludes intra-electron message passing, solving atom systems with only one nucleus could be quite a challenge since the only nucleus could become the information bottleneck. Thus, we set the number of GNN layers to eight, which is twice the normal number. Other settings of hyperparameters are shown in Apx. E. The results in Fig. 2 demonstrate that GNWF steadily provides ground state energy estimation within the chemical accuracy across all atoms. Moreover, with the increase of the atom size, GNWF shows a clear pattern that requires more training steps.

### 4.2 SOLVING GROUND STATE ENERGY FOR MOLECULES

We then examine the proposed GNWF on molecules, more complicated than atoms. We follow the selection of molecules from von Glehn et al. (2022), with LiH (4 electrons), Li2 (6), NH3 (10), CH4 (10), CO (14), N2 (14), C2H4 (16), C4H6 (30). These are all commonly seen and well-studied molecules. Detailed bond distance and bond angle information of these molecules is provided in Apx. C. In Tab. 1, we report two classical computation methods, namely the HF-CBS and CCSD(T)-CBS (the abbreviation of complete basis set). Here, we take CCSD(T)-CBS as the most accurate approximation of the exact energy, and the errors of other methods are calculated regarding this approach. All the values of these two classical approaches are directly quoted from Pfau et al. (2020)

To better evaluate the performance of the proposed GNWF, we further introduce three baseline models that are closely related to this paper: Ferminet Pfau et al. (2020), Psiformer von Glehn et al. (2022), and Globe+Moon Gao & Günnemann (2023). In von Glehn et al. (2022), the authors provided a detailed comparison of Ferminet and Psiformer on the aforementioned molecules. Therefore, the results for Ferminet and Psiformer in Tab. 1 are directly quoted from those of "Ferminet-small" and

Table 1: **Results for molecules.** The energy error is calculated regarding CCSD(T)-CBS and is in milli-Hartree with the best in bold and the second underlined. * denotes unexpected stops in the training procedure with NaN values.

| System | | HF-CBS | CCSD(T)-CBS | Ferminet | Psiformer | Globe+Moon | GNWF(ours) |
|---|---|---|---|---|---|---|---|
| LiH | Energy (Ha) | $-7.98737$ | $-8.070696$ | $-8.07050$ | $-8.070528$ | $-2.9285^*$ | $-8.070517$ |
| | Error (mHa) | 83.326 | – | 0.196 | **0.168** | – | 0.179 |
| | #params | – | – | $683,744$ | $1,593,858$ | 13M $+$1M | $936,642$ |
| Li$_2$ | Energy (Ha) | $-14.87155$ | $-14.99507$ | $-14.99480$ | $-14.99486$ | $-11.6128^*$ | $-14.99478$ |
| | Error (mHa) | 123.52 | – | 0.27 | **0.21** | – | 0.29 |
| | #params | – | – | $700,384$ | $1,602,178$ | 13M $+$1M | $953,282$ |
| NH$_3$ | Energy (Ha) | $-56.2247$ | $-56.5644$ | $-56.56347$ | $-56.56367$ | $-10.4706^*$ | $-56.56398$ |
| | Error (mHa) | 339.7 | – | 0.93 | 0.73 | – | **0.42** |
| | #params | – | – | $741,088$ | $1,621,506$ | 13M $+$1M | $953,282$ |
| CH$_4$ | Energy (Ha) | $-40.2171$ | $-40.5150$ | $-40.51430$ | $-40.51454$ | $-23.1962^*$ | $-40.51420$ |
| | Error (mHa) | 297.9 | – | 0.7 | **0.46** | – | 0.8 |
| | #params | – | – | $744,800$ | $1,622,850$ | 13M $+$1M | $988,482$ |
| CO | Energy (Ha) | $-112.7871$ | $-113.3255$ | $-113.32354$ | $-113.32416$ | $-68.2031^*$ | $-113.32403$ |
| | Error (mHa) | 538.4 | – | 1.96 | **1.34** | – | 1.47 |
| | #params | – | – | $766,944$ | $1,635,458$ | 13M $+$1M | $1,019,842$ |
| N$_2$ | Energy (Ha) | $-108.9940$ | $-109.5425$ | $-109.54046$ | $-109.54137$ | $-27.903^*$ | $-109.54166$ |
| | Error (mHa) | 548.5 | – | 2.04 | 1.13 | – | **0.84** |
| | #params | – | – | $766,944$ | $1,635,458$ | 13M $+$1M | $1,019,482$ |
| C$_2$H$_4$ | Energy (Ha) | $-78.0705$ | $-78.5888$ | $-78.58604$ | $-78.58762$ | $-31.5741^*$ | $-78.58711$ |
| | Error (mHa) | 518.3 | – | 2.76 | **1.18** | – | 1.69 |
| | #params | – | – | $799,968$ | $1,649,922$ | 13M $+$1M | $1,040,576$ |
| C$_4$H$_6$ | Energy (Ha) | $-154.9372$ | $-155.9575$ | $-155.9263$ | $-155.94619$ | – | $-155.94421$ |
| | Error (mHa) | 1020.3 | – | 31.2 | **11.3** | – | 13.3 |
| | #params | – | – | $799,968$ | $1,649,922$ | – | $1,040,576$ |

"Psiformer-small" reported in von Glehn et al. (2022). The results for Globe+Moon are obtained with the code in [1].

To evaluate the contributions of key components within our GNWF, we conducted ablation studies on NH$_3$. The results in Apx. D confirmed that removing electron-electron (e-e) edges does not lead to a loss in accuracy, and demonstrated that all newly introduced initial features (charge center and mean field embedding) contributed to the final performance improvement.

To ensure consistency with Ferminet and Psiformer, we have matched the hyperparameters of our method with those reported in von Glehn et al. (2022). Detailed hyperparameter settings for our approach are provided in Apx E. The results in Table 1 show that GNWF can barely catch up with Psiformer while reducing the number of trainable parameters by over 36.93% and the training time by 24% (16406.073 seconds for GNWF versus 21602.780 seconds for Psiformer on C$_2$H$_4$). It is important to note that the results in Table 1 are directly taken from von Glehn et al. (2022), rather than being newly replicated. In the next section, we will demonstrate that GNWF is also compatible with Psiformer when applied to previously unreported molecular geometries. These findings suggest that a carefully designed and physically informed GNN can be more efficient than simply leveraging more powerful models like the Transformer, which are frequently employed for complicated tasks.

We also replicate the models from Gao & Günnemann (2021; 2023). However, PESNet Gao & Günnemann (2021) yields results nearly identical to those of FermiNet, while Globe+Moon Gao & Günnemann (2023) consistently encounters failures during training. As a result, we omit the PESNet results and report the smoothed energy in the code before encountering NaN values for Globe+Moon.

## 4.3 APPROXIMATING ENERGY SURFACE FOR WATER

The *ab initio* method is a perfect fit for geometry optimization for unseen molecule structures due to the unique property that solving the quantum mechanical problem of a system does not rely on empirical data. We test GNWF and Psiformer on solving the energy surface of H$_2$O. The data points across varying bond lengths are obtained by iterating over the coefficients in the set $\{0.7, 0.8, 0.9, 1, 1.1, 1.2, 1.3, 1.5\}$ times 1.8 bohr. The bond angles are selected from $\{60°, 80°, 100°, 104.5°, 110°, 120°, 150°, 180°\}$, which construct a total $8 \times 8 = 64$ different H$_2$O structures. Since the results for these structures have not previously been reported in von Glehn et al. (2022), we run the code with the provided hyperparameters (see Apx. E for details).

---

[1] https://github.com/n-gao/globe

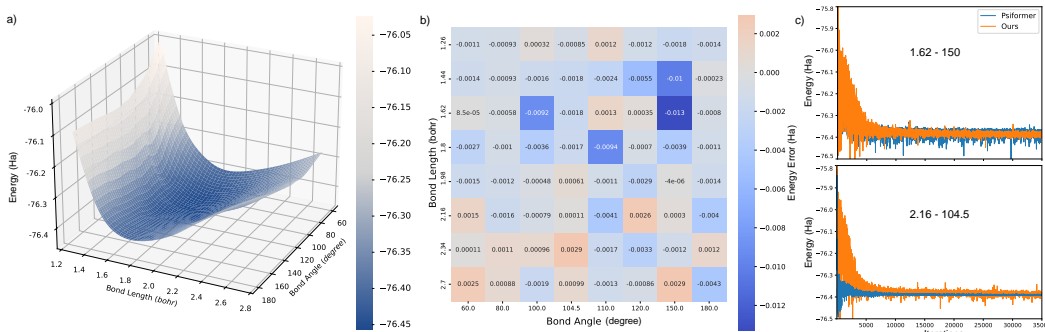

Figure 3: **Results for approximating the energy surface of water:** a) the energy surface of water with the proposed GNWF; b) the energy difference of GNWF and Psiformer, where a negative value indicates lower (better) energy for GNWF compared to Psiformer; c) examples of the training curve of both methods, with the upper one showing a relatively bad convergence of Psiformer and the lower one showing a good one.

The visualized energy surface of GNWF is provided in Fig. 3(a), which fits the expectation that the lowest energy should be around 1.8 bohr bond length and 104.5 bond angle. The energy error at the level of milli-Hartree is impossible to see in the figure, so we provide the value of the energy difference of GNWF and Psiformer in Fig. 3(b). The energy reported is $E_{GNWF} - E_{Psiformer}$, so a difference smaller than 0 indicates better results for GNWF. We also empirically find that Psiformer does not always converge as expected. Fig. 3(c) provides a comparison of the training curves of the first 35 thousand iterations. Psiformer has a much better convergence on structure 2.16 - 104.5 than 1.62 - 150, while GNWF is more steady across instances. From Fig. 3(b), we can tell that Psiformer converges better with large bond lengths and does not perform very well with small bond lengths (the training curves of all 64 instances are provided in Apx. F). This is a quite interesting case, and we suspect it is caused by the embedding initialization. Since they are concatenating the relative position of nuclei to the electron as the initial embedding, a smaller bond distance will lead to a more similar relative position for nuclei.

## 5 CONCLUSION AND OUTLOOK

As an early endeavor to machine learning at the electron level, specifically GNNs, this work has provided the so-called Graph Neural Wave Function whose computing scheme is *ab initio*. Our method and the corresponding empirical results across quantum chemistry systems showcase the ability and new potential for the intersection between machine learning and quantum science. More specifically, we set up a graph with electrons and nuclei as the nodes instead of atoms. The learned electron features are utilized as the single-particle wave function, and the many-body wave function is constructed with Slater determinant to ensure symmetries and antisymmetries. GNNs are combined with the variational Monte Carlo method to calculate the loss function and update the parameters.

Numerical results on solving the ground state energy of molecular systems illustrate the efficacy of the proposed GNWF. GNWF is able to provide accurate energy estimation within chemical accuracy for atoms and molecules and steady results across bond lengths and bond angles. The results also show that GNWF can reduce the number of trainable parameters by over 36.93% and training time by 24% compared to the SOTA approach while achieving comparable results. These findings demonstrate that a carefully designed GNN, grounded in physical intuition, can achieve competitive accuracy more efficiently than larger, more generic architectures like the Transformer. To further improve the performance of GNWF, we can develop more physically meaningful ways to construct features and pay more attention to the construction of many-body wave functions from node features.

**Limitations:** We are fully aware that the proposed method does not provide a clear margin against Psiformer in accuracy. However, we would like to point out that this might indicate the upper limits of Monte Carlo methods and Slater determinants in solving wave functions. The paper has also rethought *ab initio* NWFs by pointing out potential flaws in existing methods and discussing the difference in constructing Slater determinants, which sheds light on developing new territories for NN quantum chemistry apart from NNDFT (density functional theory).

## DECLARATION OF AI USE

We used Gemini to assist in writing:
1) Translating and polishing text in English.
All ideas, analyses, and conclusions remain our own.

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

## A    FURTHER RELATED WORKS

In this section, we provide a detailed discussion of these methods that are closely related to our work.

The introduction of neural networks to this domain was first proposed by Carleo & Troyer (2017) under the term "Neural Quantum States", where Restricted Boltzmann Machines (RBMs) were used to represent the many-body wavefunctions of spin lattice systems. Later, neural networks were applied to refine the variational ansätz through techniques such as backflow transformations Ruggeri et al. (2018); Luo & Clark (2019) acting on particle coordinates. Building on these early developments, FermiNet Pfau et al. (2020) emerged as the first NNQMC framework tailored for fermionic systems, showcasing the exceptional potential of NNQMC. Concurrently, other methods, such as PauliNet Hermann et al. (2020), achieved comparable success. More recently, transformer-based architectures like Psiformer von Glehn et al. (2022) have been developed, setting new benchmarks in performance and accuracy. In recent years, there have been many other works in the field of NNQMC, including research applied to solids Li et al. (2022b), research combined with Diffusion Monte Carlo (DMC) Wilson et al. (2021); Ren et al. (2023), improvement of existing methods Li et al. (2022a); Scherbela et al. (2022); Li et al. (2024), and applications Han et al. (2019); Hibat-Allah et al. (2020).

### A.1    GNNS FOR MOLECULAR LEARNING

GNNs Scarselli et al. (2008) provide a powerful framework for structured data representation, leveraging message-passing to capture local and global dependencies. Graph Convolutional Networks (GCNs) Kipf & Welling (2016) improved scalability by propagating node features via localized aggregation. The introduction of attention-based models, e.g. Graph Attention Network (GAT) Veličković et al. (2018), further enhanced representation learning by dynamically weighting neighboring nodes. These advancements have established GNNs as a versatile tool for complex graph-structured problems, including molecular modeling and quantum systems Yan et al. (2023); Wu et al. (2024).

Extending GNNs to respect geometric and physical constraints has led to architectures well-suited for quantum systems. SE(3)-Transformers Fuchs et al. (2020) introduce a roto-translation equivariant attention mechanism, preserving symmetry-consistent representations crucial for modeling quantum wavefunctions. E(n)-equivariant GNNs Satorras et al. (2021) further generalize these principles to arbitrary Euclidean dimensions, ensuring invariance under translations, rotations, and reflections. By embedding fundamental physical symmetries, these architectures enable more accurate and efficient quantum chemistry modeling.

GNNs have advanced molecular property prediction and structural optimization. The seminal work Duvenaud et al. (2015) introduces graph convolutions for molecular fingerprinting, later refined by Gilmer et al. (2017) with the message-passing neural network (MPNN) framework to enhance quantum property predictions, and other works Coley et al. (2019); Louis et al. (2020); Tavakoli et al. (2022). Beyond prediction, GNNs is also applied to molecular generation and optimization. Jin et al. (2018) develop the Junction Tree Variational Autoencoder (JT-VAE), enforcing chemical validity via hierarchical constraints. Reinforcement learning (RL)-based approaches, e.g. You et al. (2018), further optimize molecular design for targeted properties.

GNNs also contribute to quantum chemistry computations Reiser et al. (2022). Unke & Meuwly (2019) develop PhysNet, which efficiently approximates quantum mechanical interactions, reducing the cost of traditional DFT (density functional theory) methods. Similarly, Hamiltonian learning with GNNs has been explored for modeling electronic interactions and energy surfaces. Schütt et al. (2018) introduces a deep-learning framework for quantum Hamiltonian prediction, while Pfau et al. (2020) proposes the so-called FermiNet, which employs fully connected neural networks to represent many-electron wavefunctions, highlighting the potential of GNN-inspired architectures for electronic structure calculations. Liu et al. (2021) introduce SphereNet, a framework for learning representations of 3D molecular graphs, which significantly improves molecular property prediction and offers methods for computational chemistry. Later, a quantum embedding circuit is developed in Yan et al. (2023) to encode the 3D molecule information.

## A.2 FERMINET

The Fermionic Neural Network (FermiNet) Pfau et al. (2020) represents a significant advancement in quantum chemistry, leveraging deep learning to enhance the accuracy of variational quantum Monte Carlo (VMC) methods. By employing a neural network-based wave function Ansatz, FermiNet overcomes the limitations associated with traditional Slater-Jastrow wave functions and avoids the errors inherent in finite basis set approximations. Its novel architecture enables a compact and flexible representation of electron correlation, thereby improving the accuracy of many-electron wave function approximations.

FermiNet's architecture is designed to preserve the fundamental antisymmetry of fermionic wave functions while maintaining high expressivity. It incorporates both one-electron and two-electron features as inputs, allowing for a comprehensive representation of electronic interactions. The network employs multiple layers with distinct streams for single-electron and electron-pair features, ensuring that the wave function Ansatz captures both local and long-range correlations. Unlike traditional Slater determinants, which approximate many-electron wave functions using linear combinations of orbitals Foulkes et al. (2001), FermiNet utilizes a deep neural network to model orbitals as permutation-equivariant functions of all electron positions. This approach enhances its ability to represent complex electron correlations beyond conventional variational methods.

One of the primary advantages of FermiNet is its ability to achieve higher accuracy compared to traditional VMC approaches while remaining computationally feasible. It outperforms conventional Slater-Jastrow and Slater-Jastrow-backflow wave functions in predicting ground-state energies and demonstrates superior performance even in systems where CCSD(T) encounters difficulties.

The versatility of FermiNet is evident in its application across various atomic and molecular systems. It has been successfully employed to compute ground-state energies of first-row atoms and small molecules with near-exact accuracy. Its ability to generalize across diverse chemical environments without requiring system-specific tuning underscores its potential as a robust computational tool for electronic structure calculations.

Despite its strengths, FermiNet exhibits certain limitations. The computational cost associated with training the neural network is substantial, with the scaling of determinant evaluations reaching $\mathcal{O}(N^4)$ for large systems. Optimization complexity remains a challenge, as the stochastic nature of variational Monte Carlo requires extensive sampling to achieve convergence. Additionally, while the network architecture is flexible, achieving optimal performance for different chemical systems may necessitate hyperparameter tuning. The inclusion of a large number of determinants does not always lead to improved accuracy, suggesting that further refinement of network structures may be necessary to enhance efficiency.

## A.3 PSIFORMER

Building upon the foundation established by FermiNet, PsiFormer von Glehn et al. (2022) introduces self-attention mechanisms to improve the representation of electron correlations. While FermiNet effectively captures antisymmetric wave functions, it lacks an explicit mechanism for adaptively learning complex long-range interactions. PsiFormer addresses this limitation by integrating transformer-based multihead self-attention layers, which allow for a more dynamic and scalable treatment of electron-electron dependencies. This innovation significantly enhances its ability to model the intricate quantum mechanical correlations inherent in many-electron systems.

PsiFormer employs a single stream of self-attention layers that process electron-nuclear interactions while incorporating electron-electron effects through a Jastrow factor. This streamlined architecture enables a more efficient and expressive representation of the wave function, reducing redundant computations and improving overall accuracy. The use of self-attention further enhances scalability, allowing PsiFormer to achieve superior performance on larger molecular systems where conventional methods struggle.

PsiFormer demonstrates remarkable improvements in ground-state energy calculations compared to FermiNet, particularly for larger molecules and complex electronic environments. Its ability to achieve chemical accuracy across diverse benchmark systems highlights its potential as a transformative tool in quantum chemistry. The model not only outperforms FermiNet in absolute energy predictions but also exhibits superior generalization across different chemical environments without extensive parameter

tuning. Moreover, its self-attention framework facilitates a more interpretable representation of electronic interactions, offering deeper insights into fundamental quantum phenomena.

However, PsiFormer presents certain challenges. The computational cost associated with self-attention layers, though more efficient than traditional determinant-based methods, remains nontrivial. Additionally, the absence of direct electron-electron feature encoding within the self-attention mechanism necessitates reliance on auxiliary terms, such as the Jastrow factor, to enforce physical constraints. Future research should focus on optimizing the trade-off between accuracy and computational efficiency, as well as exploring hybrid approaches that integrate self-attention with alternative neural network architectures.

### A.4 PESNET AND BEYOND

In building upon the approaches discussed in earlier sections, the works PESNet, PESNet++, and Globe+Moon represent significant advancements in improving the scalability, speed, and versatility of neural wave function methods for solving the Schrödinger equation for many-electron systems. These models share a common foundation in the use of Graph Neural Networks (GNNs), to reparametrize and optimize wave functions across different geometries and molecular systems.

The first key advancement is PESNet, introduced by Gao & Günnemann (2021). PESNet sought to overcome the computational bottleneck of solving Schrödinger equations for different molecular geometries by reparametrizing the wave function using a meta-Graph Neural Network (MetaGNN) in conjunction with a wave function model (WFModel, basically the same with FermiNet). The architecture allows for training a single wave function that generalizes across different molecular configurations, thus significantly reducing the time required for training. PESNet was capable of capturing continuous subsets of the potential energy surface (PES) for different geometries with a single training pass, effectively enabling the modeling of energy surfaces for multiple configurations simultaneously without the need for retraining for each new geometry. Despite its advantages, PESNet still required Monte Carlo integration to estimate energies, which imposed a limitation in terms of inference speed.

To address this, PESNet++ Gao & Günnemann (2022) was introduced as an extension of PESNet. The central improvement in PESNet++ was the introduction of a surrogate model alongside the neural wave function. This surrogate model, trained using noisy energy values obtained during the VMC optimization process, allowed for sampling-free inference. By using this surrogate, PESNet++ was able to bypass the expensive Monte Carlo integration at inference time, accelerating energy evaluations from hours to milliseconds. In addition, PESNet++ incorporated several architectural improvements, such as restricted neural wave functions for closed-shell systems, leading to more accurate energy estimates, especially in challenging molecular systems. These improvements allowed PESNet++ to achieve energy errors up to $74\%$ lower than PESNet on specific benchmark molecules, making it a significant step forward in terms of both accuracy and efficiency. However, the introduction of the surrogate model means that PESNet++ can no longer comply with the first-principle calculation, which seriously reduces the physical credibility of the VMC results; moreover, the proxy model also requires additional training, which increases the additional computational overhead.

The latest one, the Globe+Moon Gao & Günnemann (2023) framework presented a novel approach to generalizing neural wave functions across different molecules. The Globe method reparametrized the wave function to accommodate molecules with varying numbers of atoms and electrons. By using localized molecular orbital embeddings and spatial message-passing in a graph network, Globe efficiently adapted neural wave functions to arbitrary molecules. The Moon model, a size-consistent neural wave function, was introduced to address the challenge of scaling wave functions to larger molecular systems. Moon, by focusing on local interactions and avoiding global interactions between distant particles, accelerated convergence in joint training and exhibited superior size consistency. Unlike previous methods, Moon demonstrated additive scaling with system size, enabling the solution of Schrödinger equations for a diverse range of molecular systems with a single, scalable wave function. However, although Moon uses GNN in its architecture, it only uses it to generalize between different systems, and does not fully utilize the advantages of GNN in efficient learning of molecules. In addition, there are problems with training stability in actual testing. Furthermore, none of these papers demonstrates the performance of their models at the chemical precision (mHa level), so their usefulness at the quantum chemical level is questionable.

## B    Further Preliminaries

This section provides additional theoretical background relevant to the study of quantum systems, focusing on the variational Monte Carlo (VMC) method, fermionic constraints, and the use of neural network methodologies in quantum mechanics.

### B.1    More about Variational Monte Carlo

The Variational Monte Carlo (VMC) method is a computational technique used to evaluate the expectation value of the Hamiltonian $\hat{H}$ through Monte Carlo sampling. The theoretical foundation of VMC rests on the variational principle, which states that for any trial quantum state $|\psi\rangle$, the energy expectation value satisfies:

$$\langle \hat{H} \rangle = \frac{\langle \psi | \hat{H} | \psi \rangle}{\langle \psi | \psi \rangle} \geq E_0, \tag{18}$$

where $E_0$ is the ground-state energy. The trial wavefunction $|\psi\rangle$ can be expanded in terms of the system's complete orthonormal eigenstates $\{|\psi_n\rangle\}$ of the Hamiltonian $\hat{H}$:

$$|\psi\rangle = \sum_n c_n |\psi_n\rangle, \tag{19}$$

where each $|\psi_n\rangle$ satisfies the stationary state Schrödinger equation $\hat{H} |\psi_n\rangle = E_n |\psi_n\rangle$. The expectation of the Hamiltonian with respect to $|\psi\rangle$ is then:

$$\langle \hat{H} \rangle = \left\langle \sum_m c_m \psi_m \middle| \hat{H} \sum_n c_n \psi_n \right\rangle = \sum_m \sum_n c_m^* c_n E_n \langle \psi_m | \psi_n \rangle = \sum_n |c_n|^2 E_n \geq E_0. \tag{20}$$

In the VMC method, the energy expectation value serves as the loss function:

$$\mathcal{L}(\theta) = \frac{\langle \psi_\theta | \hat{H} | \psi_\theta \rangle}{\langle \psi_\theta | \psi_\theta \rangle} = \frac{\int d\mathbf{r} \psi_\theta^*(\mathbf{r}) \hat{H} \psi_\theta(\mathbf{r})}{\int d\mathbf{r} \psi_\theta^*(\mathbf{r}) \psi_\theta(\mathbf{r})} = \frac{\int d\mathbf{r} |\psi_\theta(\mathbf{r})|^2 \psi_\theta^{-1}(\mathbf{r}) \hat{H} \psi_\theta(\mathbf{r})}{\int d\mathbf{r} |\psi_\theta(\mathbf{r})|^2} = \mathbb{E}_{\mathbf{r} \sim |\psi_\theta(\mathbf{r})|^2} \left[ E_L(\mathbf{r}) \right], \tag{21}$$

where the local energy is defined as:

$$E_L(\mathbf{r}) = \frac{\hat{H} \psi_\theta(\mathbf{r})}{\psi_\theta(\mathbf{r})}. \tag{22}$$

The optimization process involves iterative updates of the parameters $\theta$ using the gradient of the loss function:

$$\theta_{t+1} = \theta_t - \lambda \nabla_\theta \mathcal{L}(\theta_t), \tag{23}$$

where $\lambda$ is the learning rate. Convergence is achieved when the loss $\mathcal{L}(\theta)$ reaches a minimum, and its variance approaches zero, yielding the ground-state wave function $\psi_\theta(\mathbf{r})$. The gradient calculation of the loss function $\mathcal{L}(\theta)$ is based on some simple mathematical analysis techniques. For any variable $x$ and function $\psi(x)$, the following equations is always right:

$$\frac{\partial \log |\psi|}{\partial x} = \frac{\partial \psi}{\psi \partial x}, \quad \frac{\partial^2 \log |\psi|}{\partial x^2} = \frac{\partial}{\partial x} \left( \frac{\partial \psi}{\psi \partial x} \right) = -\frac{1}{\psi^2} \left( \frac{\partial \psi}{\partial x} \right)^2 + \frac{1}{\psi} \frac{\partial^2 \psi}{\partial x^2} = -\left( \frac{\partial \log |\psi|}{\partial x} \right)^2 + \frac{1}{\psi} \frac{\partial^2 \psi}{\partial x^2}. \tag{24}$$

In Cartesian coordinate system, using lower indices for atom $i = 1, \cdots, n$ and upper indices for the 3 components $\alpha = x, y, z$,

$$\mathbf{r}_i^\alpha \equiv \{r_i^x, r_i^y, r_i^z\} \tag{25}$$

Then the laplacian operator $\nabla_i^2$ is:

$$\nabla_i^2 = \sum_\alpha \frac{\partial^2}{\partial (r_i^\alpha)^2} = \frac{\partial^2}{\partial (r_i^x)^2} + \frac{\partial^2}{\partial (r_i^y)^2} + \frac{\partial^2}{\partial (r_i^z)^2} \tag{26}$$

The local energy:

$$E_L = \frac{1}{\psi}\hat{H}\psi = \frac{1}{\psi}\left(-\frac{1}{2}\sum_{i,\alpha}\frac{\partial^2}{\partial(r_i^\alpha)^2} + V(\mathbf{r})\right)\psi = -\frac{1}{2}\sum_{i,\alpha}\frac{1}{\psi}\frac{\partial^2\psi}{\partial(r_i^\alpha)^2} + V(\mathbf{r})$$

$$= -\frac{1}{2}\sum_{i,\alpha}\left[\frac{\partial^2\log|\psi|}{\partial(r_i^\alpha)^2}\bigg|_{\mathbf{r}} + \left(\frac{\partial\log|\psi|}{\partial r_i^\alpha}\right)^2\bigg|_{\mathbf{r}}\right] + V(\mathbf{r}).$$

$$(27)$$

As a result, for the wavefunctions, actually we only need to calculate and pass the logarithm of the absolute value of wavefunctions.

### B.2 FERMIONIC CONSTRAINTS AND SLATER DETERMINANTS

In the context of first-quantization problems in real space, our primary focus is on atomic and molecular systems. The system considered in this section is characterized by the Hamiltonian $\hat{H}$ (Equation 3). The Born-Oppenheimer approximation is employed, wherein the nuclear coordinates are assumed to be fixed while the electronic coordinates remain dynamic. Furthermore, atomic units are adopted, with energy expressed in Hartrees.

Let $N$ denote the total number of electrons in the system. The objective is to solve the stationary Schrödinger equation, $\hat{H}\psi = E\psi$, and determine the eigenfunction $\psi = \psi(\mathbf{r}) : \mathbb{R}^{3N} \to \mathbb{C}$.

For a Hermitian Hamiltonian $\hat{H}$, the eigenfunctions are real; hence, the wavefunction can be expressed as $\psi : \mathbb{R}^{3N} \to \mathbb{R}$.

In an electronic system where the spin quantum number is $n_s = \frac{1}{2}$, the spin angular momentum has two possible eigenvalues (projections), $S = \pm\frac{\hbar}{2}$, with the corresponding states denoted by $\sigma_i \in \{\uparrow, \downarrow\}$. The spatial coordinate $\mathbf{r}_i$ and spin $\sigma_i$ of electron $i$ ($i = 1, \ldots, N$) are collectively represented as $\mathbf{r}_i \triangleq \{\mathbf{r}_i; \sigma_i\}$. For brevity, the full set of electronic coordinates is written as $\mathbf{r} = \{\mathbf{r}_1, \ldots, \mathbf{r}_N\}$, leading to the wavefunction representation $\psi = \psi(\mathbf{r})$.

The antisymmetric nature of fermionic systems imposes specific constraints on the wavefunction. These constraints include:

1. **Antisymmetry:** According to the Pauli Exclusion Principle, the wavefunction changes sign upon exchanging the spin and coordinates of any two particles:

$$\psi(\ldots, \mathbf{r}_i, \ldots, \mathbf{r}_j, \ldots) = -\psi(\ldots, \mathbf{r}_j, \ldots, \mathbf{r}_i, \ldots).$$

$$(28)$$

2. **Size Consistency:** If two subsystems do not interact (or interact negligibly at large distances), the system wavefunction should be expressible as the product of the wavefunctions of the individual subsystems. For non-interacting subsystems, the wavefunction factorizes as:

$$\psi(\mathbf{r}_1, \ldots, \mathbf{r}_N) = \psi_A(\mathbf{r}_A)\psi_B(\mathbf{r}_B).$$

$$(29)$$

A widely used approach to satisfy these constraints is the Slater determinant Foulkes et al. (2001):

$$D(\mathbf{r}) = \begin{vmatrix} \phi_1(\mathbf{r}_1) & \phi_1(\mathbf{r}_2) & \cdots & \phi_1(\mathbf{r}_N) \\ \phi_2(\mathbf{r}_1) & \phi_2(\mathbf{r}_2) & \cdots & \phi_2(\mathbf{r}_N) \\ \vdots & \vdots & \ddots & \vdots \\ \phi_N(\mathbf{r}_1) & \phi_N(\mathbf{r}_2) & \cdots & \phi_N(\mathbf{r}_N) \end{vmatrix},$$

$$(30)$$

where $\phi_i(\mathbf{r}_j)$ represents a single-particle orbital, often expressed as a product of spatial and spin functions, e.g.,

$$\phi_j(\mathbf{r}_i) = \phi_j(\mathbf{r}_i)\delta_{\sigma_i\sigma_j}.$$

$$(31)$$

The Slater determinant satisfies the antisymmetry constraint naturally because swapping any two rows of a determinant reverses its sign. It also ensures equivariance since the choice of index set does not affect the determinant. For a system composed of two independent subsystems, the Slater determinant can be rearranged to be block-diagonal, satisfying size consistency.

The single-particle orbitals $\phi_j(\mathbf{r}_i)$ are obtained by solving the self-consistent Hartree-Fock (HF) equations Foulkes et al. (2001):

$$\epsilon_j \phi_j(\mathbf{r}_i) = -\left(\frac{1}{2}\nabla_i^2 + \sum_I \frac{Z_I}{|\mathbf{r}_i - \mathbf{R}_I|}\right)\phi_j(\mathbf{r}_i) + \sum_k \int d\mathbf{r}'_i \frac{|\phi_k(\mathbf{r}'_i)|^2}{|\mathbf{r}_i - \mathbf{r}'_i|}\phi_j(\mathbf{r}_i) - \sum_k \delta_{\sigma_j,\sigma_k}\int d\mathbf{r}'_i \frac{\phi_k^*(\mathbf{r}'_i)\phi_j(\mathbf{r}'_i)}{|\mathbf{r}_i - \mathbf{r}'_i|}\phi_k(\mathbf{r}_i),$$
(32)

where the Lagrange mulipliers $\epsilon_j$ comes from the orthonormality of single-particle orbitals. The first term in Eqn. (32) describes the electrons kinetic energy and electron-nucleus interactions, while the last two terms known as the Hartree and exchange terms describe the tricky electron-electron interactions.

The Slater-Jastrow wavefunction is constructed based on the Slater determinant and is expressed as Foulkes et al. (2001)

$$\psi(\mathbf{r}) = e^{J(\mathbf{r})}D(\mathbf{r}),$$
(33)

where $D(\mathbf{r})$ is given in Equation (30). The Jastrow factor, which incorporates one-body and two-body correlation terms, is defined as

$$J(\mathbf{r}) = \sum_i \chi(\mathbf{r}_i) + \sum_{i,j} u(\mathbf{r}_i, \mathbf{r}_j),$$
(34)

where the function $\chi(\mathbf{r}_i)$ accounts for nuclear-electron correlations, while $u(\mathbf{r}_i, \mathbf{r}_j)$ describes electron-electron interactions.

## C  PARAMETERS

| Atom | Position |
|------|----------|
| Li | $(-1.54957, 0.0, 0.0)$ |
| H | $(1.54957, 0.0, 0.0)$ |

Table 2: Atomic positions of LiH in Bohr radius.

| Atom | Position |
|------|----------|
| Li1 | $(-2.62015, 0.0, 0.0)$ |
| Li2 | $(2.62015, 0.0, 0.0)$ |

Table 3: Atomic positions of Li$_2$ in Bohr radius.

| Atom | Position |
|------|----------|
| N | $(0.0, 0.0, 0.22013)$ |
| H1 | $(0.0, 1.77583, -0.51364)$ |
| H2 | $(1.53791, -0.88791, -0.51364)$ |
| H3 | $(-1.53791, -0.88791, -0.51364)$ |

Table 4: Atomic positions of NH$_3$ in Bohr radius.

| Atom | Position |
|------|----------|
| C | $(0.0, 0.0, 0.0)$ |
| H1 | $(1.18886, 1.18886, 1.18886)$ |
| H2 | $(-1.18886, -1.18886, 1.18886)$ |
| H3 | $(1.18886, -1.18886, -1.18886)$ |
| H4 | $(-1.18886, 1.18886, -1.18886)$ |

Table 5: Atomic positions of CH$_4$ in Bohr radius.

| Atom | Position |
|---|---|
| C | $(0.0, 0.0, 0.0)$ |
| O | $(0.0, 0.0, 2.17381)$ |

Table 6: Atomic positions of CO in Bohr radius.

| Atom | Position |
|---|---|
| N1 | $(0.0, 0.0, 0.0)$ |
| N2 | $(0.0, 0.0, 2.13534)$ |

Table 7: Atomic positions of $N_2$ in Bohr radius.

| Atom | Position |
|---|---|
| O | $(0.0, 0.0, 0.0)$ |
| H1 | $(1.431434, 1.08334, 0.0)$ |
| H2 | $(-1.431434, 1.08334, 0.0)$ |

Table 8: Atomic positions of $H_2O$ in Bohr radius.

| Atom | Position |
|---|---|
| C1 | $(0.0, 0.0, 1.26135)$ |
| C2 | $(0.0, 0.0, -1.26135)$ |
| H1 | $(0.0, 1.74390, 2.33889)$ |
| H2 | $(0.0, -1.74390, 2.33889)$ |
| H3 | $(0.0, 1.74390, -2.33889)$ |
| H4 | $(0.0, -1.74390, -2.33889)$ |

Table 9: Atomic positions of $C_2H_4$ in Bohr radius.

| Atom | Position |
|---|---|
| C1 | $(0.0, 2.13792, 0.58661)$ |
| C2 | $(0.0, -2.13792, 0.58661)$ |
| C3 | $(1.41342, 0.0, -0.58924)$ |
| C4 | $(-1.41342, 0.0, -0.58924)$ |
| H1 | $(0.0, 2.33765, 2.64110)$ |
| H2 | $(0.0, 3.92566, -0.43023)$ |
| H3 | $(0.0, 2.33765, 2.64110)$ |
| H4 | $(0.0, -3.92566, -0.43023)$ |
| H5 | $(2.67285, 0.0, -2.19514)$ |
| H6 | $(-2.67285, 0.0, -2.19514)$ |

Table 10: Atomic positions of $C_4H_6$ in Bohr radius.

## D    MODEL ABLATIONS

To evaluate the contributions of key components within our neural wave function, we conducted ablation studies on the $NH_3$ molecule. We conducted tests where we added electron-electron message passing by increasing the number of edges to form a fully connected graph. However, we found that this modification led to poorer convergence compared to our proposed method. The model variant utilizing only electron-electron message passing yielded an energy of -56.544006, in contrast to the -56.56398 achieved by the full GNWF model. Furthermore, ablating the charge-cen feature or the mean-field embeddings resulted in energies of -56.52344 and -56.26201, respectively. These components were ablated by setting their corresponding input dimensions to unity for all particles.

|  | Result Energy (Ha) | Error (mHa) |
|---|---|---|
| Proposed GNWF | -56.56398 | 0.42 |
| With e-e message passing | -56.54406 | 20.34 |
| Without charge-cen | -56.52344 | 40.96 |
| Without mean-filed embeddings | -56.26201 | 302.39 |

Table 11: Model ablations tests

## E    HYPERPARAMETERS

|  | Hyperparameter | Value |
|---|---|---|
| Pretraing | Steps | 2e4 |
|  | Basis Set | STO-6G |
|  | Method | RHF |
| Optimization | Steps | 5e4 |
|  | Optimizaer | kfac |
|  | Learning rate | $\frac{0.05}{1+\frac{t}{1e5}}$ |
|  | Batch size | 4096 |
|  | Local energy clipping | 5 |
| Psiformer | Layers | 4 |
|  | Heads | 4 |
|  | Head dim | 64 |
|  | MLP hidden dim | 256 |
|  | Use layer norm | True |
| MCMC | Target pmove | 0.5 |
|  | # Steps | 10 |

Table 12: Hyperparameters of Psiformer

The hyperparameters for Psiformer, Globe and our model are presented above. For Psiformer and Globe, we directly utilize the hyperparameters given in Psiformer von Glehn et al. (2022) and Globe+Moon Gao & Günnemann (2023), while in Posiformer we reduce the optimization steps to 5e4 due to limited computational resources. For our model, we adopt the same optimizer as Psiformer, which demonstrates better convergence time and stability in our experiments. The number of optimization steps of our model varias across three different experiments in the main text. The first two experiments use 1e5 steps each to run throughout the variational optimization process , whereas the third one takes 5e4 steps for each geometry of $H_2O$ to conduct a comparison experiment with Posiformer.

|  | Hyperparameter | Value |
|---|---|---|
| Pretraing | Steps | 2e4 |
|  | Basis Set | STO-6G |
|  | Method | RHF |
| Optimization | Steps | 5e4 or 1e5 |
|  | Optimizaer | kfac |
|  | Learning rate | $\frac{0.1}{1+\frac{t}{1e4}}$ |
|  | Batch size | 4096 |
|  | Local energy clipping | 5 |
| GNN | Layers | 4 |
|  | Edge feature dim | 64 |
|  | Node feature dim | 256 |
| MCMC | Target pmove | 0.5 |
|  | # Steps | 10 |

Table 13: Hyperparameters of Ours

|  | Hyperparameter | Value |
|---|---|---|
| Pretraing | Steps | 1e4 |
|  | Basis Set | STO-6G |
|  | Method | RHF |
| Optimization | Steps | 6e4 |
|  | Learning rate | $\frac{0.1}{1+\frac{t}{100}}$ |
|  | Batch size | 4096 |
|  | Damping | 1e-4$\sigma[E_L]$ |
|  | Local energy clipping | 5 |
|  | Max grad norm | 1 |
|  | CG max steps | 100 |
| MCMC | Target pmove | 0.5 |
|  | # Steps | 40 |
| Moon | Hidden dim | 256 |
|  | E-E int dim | 32 |
|  | Layers | 4 |
|  | Activation | SiLU |
|  | Determinants | 16 |
|  | Jastrow layers | 3 |
|  | Filter hidden dims | [16,8] |
| Reparameterization | Embedding dim | 128 |
|  | MLP layers | 4 |
|  | Message dim | 64 |
|  | Layers | 3 |
|  | Activation | SiLU |
|  | Filter hidden dims | [64,16] |

Table 14: Hyperparameters of Globe

# F  OPTIMIZATION PROCESS OF PSIFORMER AND OUR MODEL

This section shows the training process of the models.

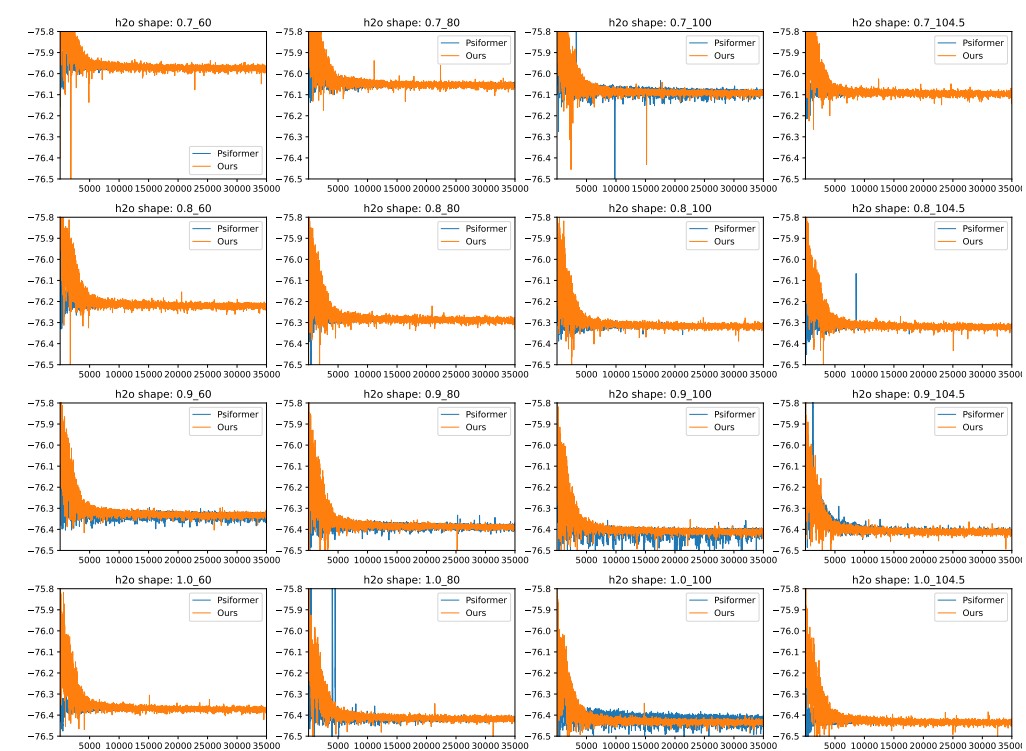

Figure 4: The optimization process of Psiformer and our model is shown. The geometries of $H_2O$ molecule are displayed above each subplot, where the first factor indicates the relative bond length to the optimal geometry, and the second refers to bond angle measured in degrees.

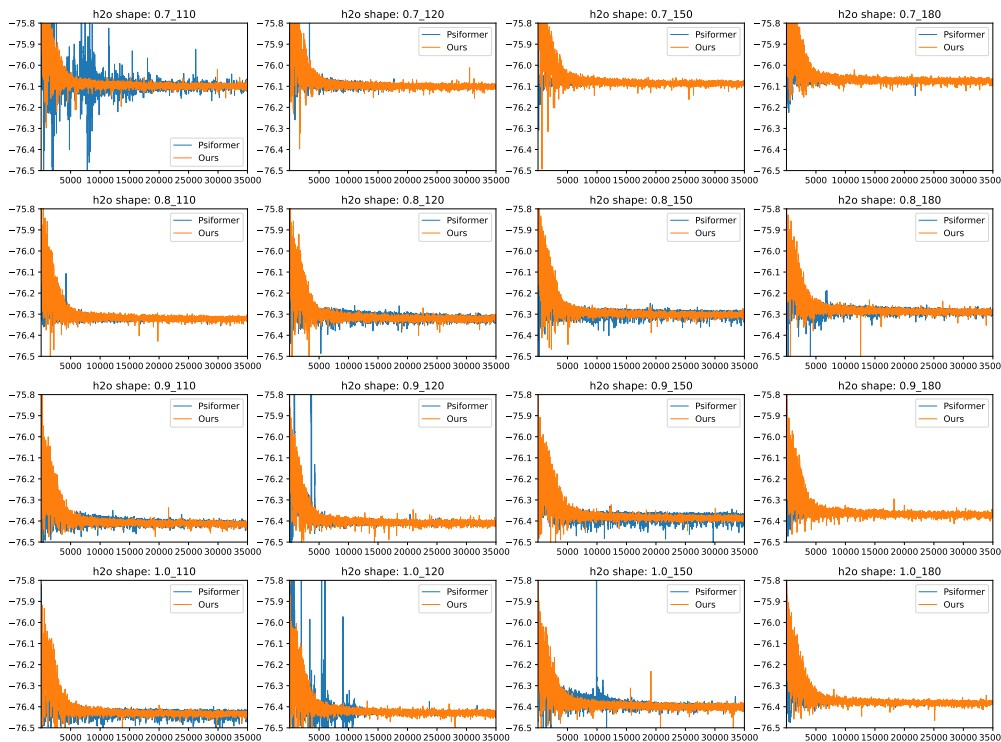

Figure 5: The optimization process of Psiformer and our model is shown. The geometries of $H_2O$ molecule are displayed above each subplot, where the first factor indicates the relative bond length to the optimal geometry, and the second refers to bond angle measured in degrees.

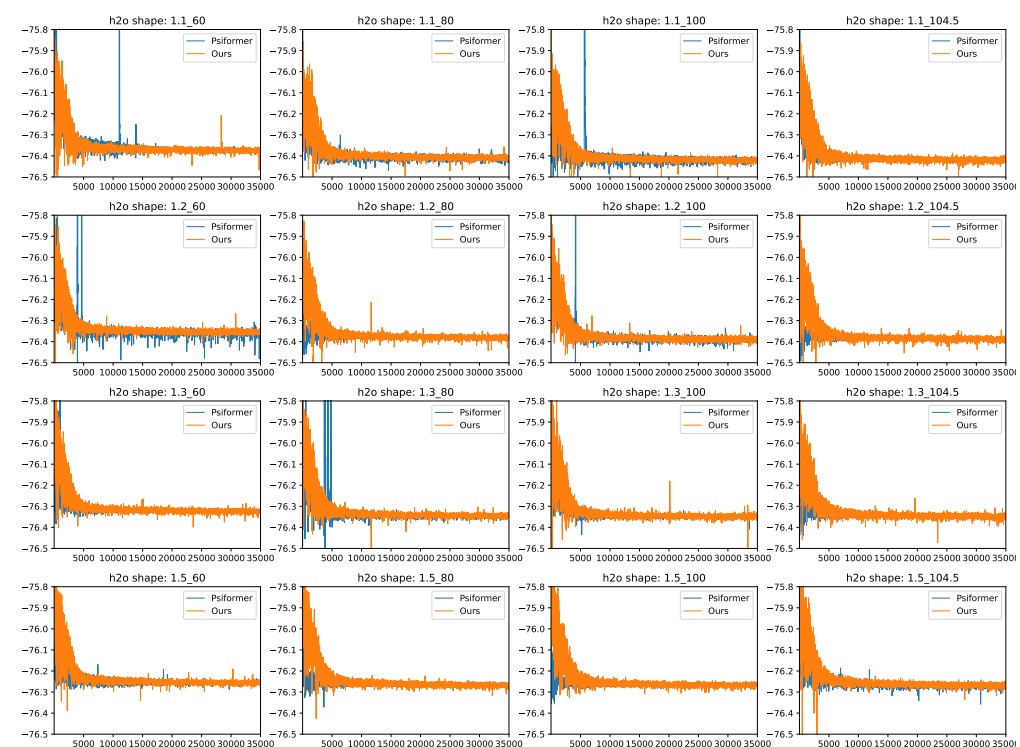

Figure 6: The optimization process of Psiformer and our model is shown. The geometries of $H_2O$ molecule are displayed above each subplot, where the first factor indicates the relative bond length to the optimal geometry, and the second refers to bond angle measured in degrees.

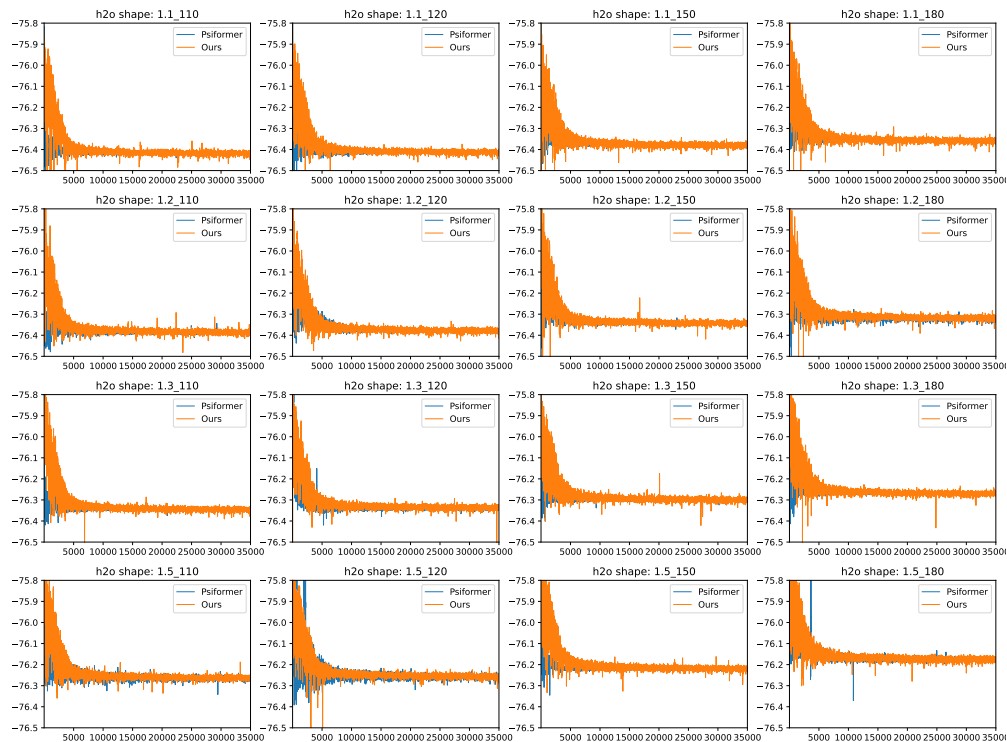

Figure 7: The optimization process of Psiformer and our model is shown. The geometries of $H_2O$ molecule are displayed above each subplot, where the first factor indicates the relative bond length to the optimal geometry, and the second refers to bond angle measured in degrees.

