# OpenReview forum: "Rethinking ab initio Neural Wave Function: GNNs at Electron Level"
_ICLR.cc/2026/Conference — Submitted to ICLR 2026_

### Official Review · Reviewer_dZTs · 2025-10-24

**Soundness:** 1
**Presentation:** 2
**Contribution:** 1
**Rating:** 2
**Confidence:** 5

**Summary:**

This work introduces a graph neural network-based wave function ansatz for solving the time-independent many-body Schrödinger equation. The authors argue that explicit electron-electron correlations within the Ansatz hurt the model and instead present a message passing neural network that passes information from electrons to nuclei and vice versa. In their experiments, the authors find a speedup of 24% compared to transformer-based wave functions.

**Strengths:**

The text and figures are clear and legible.

**Weaknesses:**

The paper has several substantial weaknesses that, in its current form, preclude publication.

# Related work
The discussion of related work is incomplete and lacks sufficient depth. Several key neural network wave function (NNWF) and graph neural network (GNN)-based architectures are missing or only superficially discussed:
* PauliNet [1], one of the first NNWFs, was heavily inspired by SchNet but incorporates explicit electron–electron message passing.
* FiRE [3] is a recent GNN-based wave function that achieves over 10× faster variational Monte Carlo (VMC) simulations by exploiting sparsity patterns in message passing.
* Moon [2] is conceptually very similar to the proposed GNWF, with the main difference being the absence of the initial electron–electron message-passing step.
* [4] extends FermiNet by applying SchNet-like filters to electron–electron messages.
* Contrary to the claim on l.212, PsiFormer [5] does not decouple electron–electron interactions into a Jastrow factor but rather models them explicitly via self-attention. The Jastrow term only contains explicit pairwise coordinates to satisfy Kato’s cusp condition.
* Works on generalized wave functions such as PESNet or Globe remain ab initio methods, contrary to the claim in l.146–148. The meta-network in these models forms part of the wave function, even though it operates on nuclear positions.

# Theory
While it is correct that nuclei form the primary centers of electronic interaction (l.207–210), this framing does not accurately capture the main problem that NNWFs aim to address. The mean-field energy contribution is already well described by Hartree–Fock theory. NNWFs instead target the residual correlation energy—a small but high-frequency component arising from electron–electron correlations. This aspect is essential and is the primary motivation for the complex electron–electron interaction terms in NNWF architectures.


# Experiments
The experimental results are unconvincing and show inconsistencies as well as questionable interpretations:
* Figure 2: The original FermiNet paper [7] reported substantially lower errors. In [6], the authors trained a single wave function (rather than eight separate ones) across all atoms and their ionizations, achieving higher accuracy on all systems. For single-structure training, even higher accuracy would thus be expected [7].
* Table 1: CCSD(T)-CBS does not provide reliable absolute energies and should only be used for estimating relative energies. As demonstrated in [6], larger systems can yield considerably lower absolute energies. The inclusion Globe as baselines is unnecessary in this context, as no generalization is evaluated, these numbers should be double checked and including failed runs as total energies seems misleading.
* Overall, comparisons should focus on relative rather than absolute energies.

## Minor
* l.16,19: Electrons are not inherently “graphical”; they constitute a point set.
* l.25: Clarify what new capabilities or insights are unlocked by the GNWF formulation.
* l.74: Prefer using mathematical symbols rather than “pos.”
* l.93: “Ansatz” (German for “approach”) denotes the functional form used to approximate the (ground-state) wave function.
* Figure 1: The label “Single-Particle Wave Function” is misleading, as it could be mistaken for a mean-field wave function.
* l.188–189: While translational invariance is enforced via relative coordinates, the wave function is not rotationally invariant or equivariant. True rotational invariance would require SO(3)-equivariance, but in quantum systems spontaneous symmetry breaking can occur (e.g., excited states of hydrogen).
* l.296–300: The description implies that the many-electron orbitals are a novel feature of GNWFs, but such constructs are standard in existing NNWF architectures.

[1] Hermann, Jan, Zeno Schätzle,Frank Noé. „Deep-Neural-Network Solution of the Electronic Schrödinger Equation“\
[2] Gao, Nicholas,Stephan Günnemann. „Generalizing Neural Wave Functions“.\
[3] Scherbela, Michael, Nicholas Gao, Philipp Grohs,Stephan Günnemann. „Accurate Ab-Initio Neural-Network Solutions to Large-Scale Electronic Structure Problems“\
[4] Gerard, Leon, Michael Scherbela, Philipp Marquetand,Philipp Grohs. „Gold-standard solutions to the Schrödinger equation using deep learning: How much physics do we need?“\
[5] Glehn, Ingrid von, James S. Spencer, David Pfau. „A Self-Attention Ansatz for Ab-Initio Quantum Chemistry“\
[6] Gao, Nicholas, Stephan Günnemann. „Neural Pfaffians: Solving Many Many-Electron Schrödinger Equations“\
[7] Pfau, David, James S. Spencer, Alexander G. D. G. Matthews, W. M. C. Foulkes. „Ab initio solution of the many-electron Schrödinger equation with deep neural networks“.

**Questions:**

In context of the related works, I wonder where the contribution of this work lies as message passing networks have commonly be used in NNWFs.

---

### Official Review · Reviewer_wBfK · 2025-10-25

**Soundness:** 3
**Presentation:** 3
**Contribution:** 3
**Rating:** 4
**Confidence:** 4

**Summary:**

The paper introduces **Graph Neural Wave Function (GNWF)**, a neural ansatz for ab initio electronic structure modeling. GNWF treats electrons and nuclei as nodes in a graph and performs message passing only between electron–nucleus pairs to encode interactions efficiently. The learned electron embeddings are used to form a Slater determinant, trained via Variational Monte Carlo to estimate ground-state energies. By removing electron–electron edges, GNWF achieves strong accuracy with fewer parameters and faster training than Psiformer, while maintaining stability across molecular geometries. Experiments on atoms and small molecules demonstrate *chemical accuracy*, about *24% faster convergence*, and *37% parameter reduction*, showing that GNWF offers a compact and physically grounded alternative for neural wave-function learning.

**Strengths:**

1. GNWF innovatively employs graph neural networks at the electron level, offering a physically grounded and interpretable architecture that explicitly respects quantum symmetries and distance-based interactions.
2. Experiments show GNWF achieves comparable or better accuracy than Psiformer with fewer parameters and faster convergence, demonstrating strong generalization and stability across diverse molecular configurations.

**Weaknesses:**

1. Restricting message passing to electron–nucleus pairs may limit expressiveness and omit explicit electron–electron correlation. Depite the information is merged at the end, the expressiveness might be seriously compromised. The exponential distance decay, if truncated, could introduce gradient discontinuities, and the conceptual distinction from Transformer-style global attention remains unclear.

2. Several design choices, e.g. dynamic nuclear embeddings, mean-field and center-of-charge features, and exponential decay, lack clear justification or ablation analysis, making it difficult to assess their individual contributions or necessity.

3. The empirical evaluation is confined to small molecules with weak baseline tuning and limited scalability demonstrations. Despite efficiency gains, GNWF underperforms Psiformer on key energy metrics, leaving its practical advantage uncertain.

**Questions:**

1. How well does GNWF scale to larger or periodic systems beyond the small molecules (~30 electrons) tested, and what are the computational or architectural bottlenecks that may limit its applicability?
2. Can the authors provide deeper insight into the learned representations? Additionally, why was the exponential decay function exp(−r) selected for message weighting, and were other functional forms or ablation studies considered?
3. The reported performance of GNWF is lower than Psiformer on several benchmarks, and the Globe+Moon baseline seems under-tuned. Could the authors clarify whether the observed performance gaps arise from optimization difficulties, architectural constraints, or training instability and report the difficulties with training Globe+Moon?

I am willing to raise the rating if the authors can address these concerns and questions.

---

### Official Review · Reviewer_yFEa · 2025-10-31

**Soundness:** 2
**Presentation:** 2
**Contribution:** 2
**Rating:** 2
**Confidence:** 5

**Summary:**

Solving the multi electron Schrödinger equation represents one of the most important problems in computational chemistry. In recent years, methods based on neural network parametrizations of the wavefunction have been developed and demonstrated to outperform well established sota methods. In this paper a GNN bases ansatz for fermionic neural wave functions is proposed. Following earlier work in this field, the multi electron Schrödinger equation is then solved via Variational Monte Carlo by minimizing the energy over the ansatz. The proposed architecture is tested on small molecules with up to 30 electrons. It is claimed that the method achieves comparable accuracy to methods like Psiformer while requiring fewer trainable parameters, resulting in a computational speedup of around 24%.

**Strengths:**

The topic treated in the paper is highly relevant and significant advances in neural network based VMC could lead to big breakthroughs in science (development of pharmaceuticals or materials). Therefore it is a key problem to find optimal architectures. The proposed GNN approach adds a new one to the existing choices.

**Weaknesses:**

Arguably the GNN approach is not really new, as it has been already explored in work by Gao&Guennemann, albeit in a slightly different form. While the proposed architecture is not exactly present in earlier literature, I do feel that it represents a rather incremental contribution that is not suitable for ICLR. Moreover, I am not convinced about the main claim regarding competitiveness with existing methods. The paper only reports experiments with systems of up to 30 electrons. Note that state of the art fermionic neural wave functions are now capable of treating challenging systems with several hundreds of electrons (see https://arxiv.org/abs/2505.19909 or https://arxiv.org/abs/2504.06087). But even on very small systems with 30 electrons (for example bicyclobutane), the presented architecture misses the accuracy of the "small" Psiformer model by 2mHa and of the "large" Psiformer by more than 13mHa. The crucial question whether the presented model is capable to match the accuracy of these more accurate models by increasing the parameters is not studied. For these reasons I do not think that the claim that the GNN approach yields "comparable accuracy" to Psiformer is not fully justified. Moreover, the claimed speedup of 24% is respectable but in my opinion not sufficient to justify publication in ICLR. Finally I mention that the accuracies of Psiformer are simply taken from the corresponding paper. It is entirely possible that reducing the Psiformer model a bit (fewer parameters) and running the experiments again would yield the same or better accuracy than the proposed method with even fewer parameters.

The paper seems not very carefully written. I could find several typos and repetitions. Moreover the technical part B.1/B.2 is confusing. The paper mentions size consistency but the VMC ansatz is not size consistent. Morever spin symmetry is briefly mentioned but it is not explained how it is enforced. I also did not find information on the optimizer, on how MCMC sampling is conducted, or how many determinants are used.

**Questions:**

How many Slater determinants are used?

How is Spin symmetry enforced?

Does the model, when scaled up, reach the accuracy of Psiformer (large)? If yes, does it require fewer parameters than Psiformer?

In the conclusion it is written that "The paper has also rethought ab initio NWFs by pointing out potential flaws in existing methods". Please clarify. What precisely is "rethought" and what are the "flaws" in existing methods (including Ferminet/Psiformer but also many other ones)?

---

### Official Review · Reviewer_Twfc · 2025-10-31

**Soundness:** 2
**Presentation:** 1
**Contribution:** 2
**Rating:** 2
**Confidence:** 4

**Summary:**

This paper introduces Graph Neural Wave Function (GNWF), a new neural ansatz for ab initio electronic structure modeling. Instead of using Transformer-based architectures such as PsiFormer or meta-GNN parameterizations like Globe+Moon, the authors propose to use a GNN directly as the wavefunction ansatz itself. Specifically, the GNN performs message passing over a bipartite electron–nucleus graph to generate single-particle orbitals, which are then combined via Slater determinants to ensure antisymmetry.

**Strengths:**

**Revisiting the Role of the Ansatz in Neural Wavefunctions**: The paper provides a fresh perspective on how strongly the functional form of a wavefunction should be imposed. By replacing an explicit ansatz (e.g., Transformer with pre-defined feature mixing) with a learnable GNN structure, the authors reopen an important question: whether a data-driven, graph-based representation can recover the same physical structure without explicitly handcrafting it. This is a meaningful conceptual contribution, even if not fully realized experimentally.

**Weaknesses:**

**Lack of Analysis on the Source of Efficiency**: The reported speedup and parameter reduction are not supported by sufficient analysis. It remains unclear where the efficiency arises—from algorithmic complexity (O(N·M) vs. O(N²)), parameter count reduction, or implementation details. There is no complexity ablation or equal-parameter comparison. For example, it is uncertain whether a PsiFormer with the same number of parameters as GNWF would show comparable performance. Without such fairness analysis, the claim of architectural efficiency remains speculative.

**Unconvincing Argument on Physical Grounding**: The paper claims GNWF is more “physically grounded” than Transformer- or Pfaffian-based ansatzes, but this is not logically or empirically justified. The traditional ansatz methods (Slater + Jastrow, Pfaffian, FermiNet) are themselves physically constrained and theoretically well-grounded in antisymmetry and orbital factorization. The paper does not provide a clear argument for why these methods are “less physical.” Instead, it relies solely on energy results—which are not superior to PsiFormer—to support this claim. As a result, the notion of “physically grounded” remains qualitative rather than quantitative.

**Lack of Comparison to Other Modern Ansatz Models**: The paper does not include comparisons to recent, expressive anti-symmetric architectures such as Neural Pfaffian or PauliNet, which are directly relevant baselines. These models specifically address the same challenge (expressivity and efficiency of antisymmetric ansatz), and omitting them weakens the empirical validation of GNWF’s claimed advantages.

**Questions:**

1. **Neural Pfaffian Baseline**: Why was the Neural Pfaffian not included in the comparison?
This model provides an alternative antisymmetric ansatz that directly competes with GNWF in terms of expressivity and efficiency. Moreover, since a Pfaffian-based antisymmetry can, in principle, also be incorporated into the GNWF framework (for example, by using GNN-derived single-particle orbitals as the input to a Pfaffian determinant), it would be valuable to either (a) include this variant or (b) discuss how such an integration could be achieved. Evaluating GNWF with a Pfaffian-style antisymmetry could strengthen the argument that the GNN-based ansatz is general enough to subsume existing architectures.

2. **Globe+Moon Comparison under Convergent Settings** :The paper states that Globe + Moon training repeatedly failed to converge. Could you clarify whether this failure is due to unstable optimization, mismatched hyperparameters, or architectural incompatibility with your data setup? If Globe + Moon can be made to converge under its original settings, would GNWF still outperform it in accuracy or computational efficiency on the same settings? A controlled comparison under a convergent configuration would clarify the source of differences.

3. **Fair Parameter and Runtime Comparison**: The reported runtime improvement might partly stem from using fewer parameters rather than from intrinsic architectural advantages. Could you provide results where PsiFormer and GNWF are trained under equal parameter budgets? For example, how does GNWF perform if PsiFormer is down-scaled to the same parameter count (~1.04 M)? Conversely, if GNWF uses the full parameter size of PsiFormer (~1.65 M), does its accuracy or speed change proportionally? Such experiments would make the efficiency claim substantially more convincing.

---

### Meta-Review · Area_Chair_Jhxa · 2026-01-04

**Summary:**

The paper proposes a Graph Neural Wave Function (GNWF) as an ansatz for Variational Quantum Monte Carlo (VMC) to solve the many-body Schrödinger equation. The method utilizes a graph neural network architecture defined over a bipartite graph of electrons and nuclei, notably omitting explicit electron-electron message passing edges to improve computational efficiency. The authors claim comparable accuracy to recent transformers like PsiFormer with a 24% speedup and reduced parameter count.

**Reviewer Concerns:**

**1. Incomplete Related Work and Missing Baselines**
A critical issue identified by multiple reviewers is the omission of highly relevant prior work.

* The paper fails to compare against or discuss **Neural Pfaffian**, **PauliNet**, and **FIRE**, which are direct competitors in terms of expressivity and efficiency.


* Reviewers noted that the "Moon" architecture is conceptually very similar to GNWF, yet not adequately distinguished.


* There are inaccuracies in the description of baselines; specifically, the paper incorrectly claims PsiFormer decouples electron-electron interactions into a Jastrow factor, whereas it actually models them explicitly via self-attention.



**2. Insufficient Empirical Validation**
The experimental results do not sufficiently support the claims of "comparable accuracy."

* **Accuracy Gap:** The method underperforms PsiFormer (both small and large variants) on small systems (up to 30 electrons). Reviewers noted that omitting explicit electron-electron correlations likely compromises expressiveness, leading to this performance gap.


* **Questionable Baselines:** Reviewers pointed out that the FermiNet baselines reported in the paper show significantly higher errors than those reported in the original FermiNet literature, casting doubt on the fairness of the comparison.


* **Scalability:** The experiments are limited to small molecules, whereas state-of-the-art methods are currently evaluated on systems with hundreds of electrons.



**3. Lack of Ablation and Analysis**
The source of the claimed efficiency is not well-analyzed.

* It is unclear if the speedup stems from the architectural change $(O(N)$ vs $O(N^2))$ or simply a reduction in parameters.


* Reviewers requested "equal-parameter" comparisons to determine if GNWF is truly more efficient per parameter than PsiFormer, which were not provided.


* Design choices, such as the specific exponential decay function, lack ablation studies.



**4. Theoretical Justification**
The claim that GNWF is more "physically grounded" was challenged. Reviewers argued that traditional ansatz methods (Slater-Jastrow, FermiNet) are already physically constrained and that the paper relies on qualitative arguments rather than quantitative evidence to support this superiority.

**Reviewer Scores:**

2 2 4 2

---

### Decision · Program_Chairs · 2026-01-26

Reject